# Whole genome sequences of multi-drug resistant *Escherichia coli* isolated in a Pastoralist Community of Western Uganda: Phylogenomic changes, virulence and resistant genes

Jacob Stanley Iramiot[1,2]*, Henry Kajumbula[1], Joel Bazira[3], Etienne P. de Villiers[4,5,6], Benon B. Asiimwe[1]

1 Department of Medical Microbiology, College of Health Sciences, Makerere University, Kampala, Uganda, 2 Department of Microbiology and Immunology, Faculty of Health Sciences, Busitema University, Mbale, Uganda, 3 Department of Microbiology, Faculty of Medicine, Mbarara University of Science and Technology, Mbarara, Uganda, 4 KEMRI-Wellcome Trust Research Programme, Centre for Geographic Medicine Research-Coast, Kilifi, Kenya, 5 Centre for Tropical Medicine and Global Health, Nuffield Department of Medicine Research Building, University of Oxford, Oxford, United Kingdom, 6 Department of Public Health, Pwani University, Kilifi, Kenya

* jiramiot@gmail.com

**Data Availability Statement:** Data is available in a public repository; DOI 10.17605/OSF.IO/KPHRD.

## Abstract

### Background

The crisis of antimicrobial resistance is already here with us, affecting both humans and animals alike and very soon, small cuts and surgeries will become life threatening. This study aimed at determine the whole genome sequences of multi-drug resistant *Escherichia coli* isolated in a Pastoralist Community of Western Uganda: phylogenomic changes, virulence and resistant genes.

### Methods

This was a laboratory based cross sectional study. Bacterial isolates analyzed in this study were 42 multidrug resistant *E. coli* isolated from stool samples from both humans (n = 30) and cattle (n = 12) in pastoralist communities collected between January 2018-March 2019. Most of the isolates (41/42) were resistant to three or more antibiotics (multi-drug resistant) and 21/42 isolates were ESBL producers; 13/30 from human and 8/12 from cattle. Whole Genome Sequencing (WGS) was carried out at the facilities of Kenya Medical Research Institute-Wellcome trust, Kilifi, to determine the phylogenomic changes, virulence and resistant genes.

### Results

At household level, the genomes from both human and animals clustered away from one another except for one instance where two human isolates from the same household clustered together. However, 67% of the *E. coli* isolated from cattle were closely related to those

**Funding:** This work was supported by the DELTAS Africa Initiative [grant# 107743/Z/15/Z]. The DELTAS Africa Initiative is an independent funding scheme of the African Academy of Sciences (AAS)'s Alliance for Accelerating Excellence in Science in Africa (AESA) and supported by the New Partnership for Africa's Development Planning and Coordinating Agency (NEPAD Agency) with funding from the Wellcome Trust [grant #107743/Z/15/Z] and the UK government. The views expressed in this manuscript are those of the author(s) and not necessarily those of AAS, NEPAD Agency, Wellcome Trust or the UK government.

**Competing interests:** The authors have declared that no competing interests exist.

**Abbreviations:** ESBLs, Extended spectrum β-lactamases; CTAB, Cetyltrimethylammonium bromide; DNA, Deoxyribonucleic acid; CLSI, Clinical and Laboratory Standards Institute guidelines; WGS, Whole Genome Sequencing; AMR, antimicrobial resistance; QEPA, Queen Elizabeth National Park; UNCST, Uganda National Council for Science and Technology.

found in humans. The *E. coli* isolates were assigned to eight different phylogroups: A, B1, B2, Cladel, D, E, F and G, with a majority being assigned to phylogroup A; while most of the animal isolates were assigned to phylogroup B1. The carriage of multiple AMR genes was higher from the *E. coli* population from humans than those from cattle. Among these were Beta-lactamase; blaOXA-1: Class D beta-lactamases; blaTEM-1, blaTEM-235: Beta-lactamase; catA1: chloramphenicol acetyl transferase; cmlA1: chloramphenicol efflux transporter; dfrA1, dfrA12, dfrA14, dfrA15, dfrA17, dfrA5, dfrA7, dfrA8: macrolide phosphotransferase; oqxB11: RND efflux pump conferring resistance to fluoroquinolone; qacL, qacEdelta1: quinolone efflux pump; qnrS1: quinolone resistance gene; sul1, sul2, sul3: sulfonamide resistant; tet(A), tet(B): tetracycline efflux pump. A high variation of virulence genes was registered among the *E. coli* genomes from humans than those of cattle origin.

## Conclusion

From the analysis of the core genome and phenotypic resistance, this study has demonstrated that the *E. coli* of human origin and those of cattle origin may have a common ancestry. Limited sharing of virulence genes presents a challenge to the notion that AMR in humans is as a result of antibiotic use in the farm and distorts the picture of the directionality of transmission of AMR at a human-animal interface and presents a task of exploring alternative routes of transmission of AMR.

## Background

The crisis of antimicrobial resistance is already here with us, affecting both humans and animals alike and very soon, small cuts and surgeries will become life threatening. Evidence of non-prescribed use of antimicrobials in livestock to mask ill farming practices and misuse of antibiotics in community pharmacies has been documented globally [1, 2]. Farmers use large amounts of antibiotics in livestock and this is now known as a key driver and recipe to accelerating emergency of antimicrobial resistance [3]. Resistant bacterial clones may spread from animals to humans rendering antibiotics less effective and increases mortality and morbidity in developing nations due to such bacteria [3]. Previous studies in Uganda report abuse of antimicrobials in animal husbandry as a major contributor to antimicrobial resistance emergence among microbes. Additionally, 40% of the persons who visit a health- care facility in Uganda are treated with antibiotics [4]. These antibiotics are mostly obtained without prescription in drug shops and community pharmacies in sub-therapeutic doses. Globalization of trade coupled with the revolutionalization of travel has simplified distribution of resistant bacteria due to easy movement of humans and their goods including livestock across countries making the problem of antimicrobial resistance (AMR) global in nature.

The global spread of multi-drug resistant Enterobacteriaceae especially CTX-M type ESBLs and strains producing carbapenemases such as KPC and NDM warrant a multi-stake holder attention [5]. Resistance against certain antibiotic categories is already high in hospital settings in Uganda but resistance among bacteria from animals is not well documented. Introduction of bacteria with resistant genes into food animals as a result of antibiotic selection in veterinary medicine is now of great concern. Due to the routine consumption of certain antibiotics like tetracyclines in animal husbandry, undesirable shift of resistance towards potent second line

and third generation antibiotics is now a reality [6, 7]. Baseline knowledge on the epidemiology and transmission routes of drug resistant microbes in Uganda is needed to drive the necessary prevention, intervention and control measures. While studies have been conducted in health care facilities in mostly urban areas settings [8, 9]. Little effort has been devoted to determining the molecular epidemiology of antimicrobial resistance, including multidrug resistance at a human-animal interface. Pastoralist communities live with their domestic animals inside the park hence a porous interface for microbial and disease transmission which provides a good ground for this study. We aimed to determine the whole genome sequences of multi-drug resistant *Escherichia coli* isolated in a Pastoralist Community of Western Uganda: phylogenomic changes, virulence and resistant genes.

## Methods

### Bacterial strains

Bacterial isolates analyzed in this study were multidrug resistant bacteria isolated from stool samples from both humans (n = 30) and cattle (n = 12) in pastoralist communities of Kasese district between January 2018-March 2019. The cattle were sample because they were the most common animals reared by the pastoralists in Kasese district. The pastoralist communities in Kasese district are settled in and around Queen Elizabeth National Park (QENP). The Kasese side of the National Park has two pastoralist communities in Nyakatonzi and Hima sub-countries. The QEPA lies astride the equator along the latitudes of 0Ê 39' 36" North, 30Ê 16' 30" East. The northern area has been occupied by pastoralists since the 1920s. Specific sites where samples were taken were Bwera-Mpondwe (Bwera) in the east, Hima in the north and Katwe-Kabatoro (Katwe).

### Speciation and antibiotic susceptibility testing

Speciation and antibiotic susceptibility of the isolates was done using the Phoenix automated microbiology system (Phoenix 100 ID/DST system) from Becton and Dickson (Franklin Lakes, NJ, USA) and the results interpreted using the CLSI guidelines. Sensitivity testing was carried out using a total of 15 antibiotics which include; ampicillin, amoxicillin-clavulanic acid, cefazolin, cefuroxime, ceftazidime, ceftriaxone, cefepime, ciprofloxacin, levofloxacin, gentamycin, tetracycline, nitrofurantoin, imipenem, Ertapenem and cotrimoxazole. Multi-drug resistance was defined as one isolate being resistant to three or more classes of antibiotics tested [10].

### Extraction of Genomic bacterial DNA

Extraction of Genomic bacterial DNA was done at Molecular Biology Laboratory of Department of Immunology and Molecular Biology of Makerere University using Modified Cetyltrimethylammonium bromide (CTAB) Method as described before [11]. The modified CTAB method uses Enzymes and detergents to lyse cells, release the nucleic acids, and an organic solvent to purify the DNA and absolute alcohol (isopropanol/ethanol) to precipitate out the DNA.

### Whole genome-sequencing

Whole Genome Sequencing (WGS) was carried out at the facilities of KEMRI Wellcome Trust Research Programme for the isolates that turned out to be multidrug resistant from house contacts and cattle. Genomic DNA from cultured *E. coli* was quantified using Qubit and diluted to 0.2ng/μl and library prep performed using the Illumina Nextera XT protocol according to the

manufactures instructions. Briefly, gDNA was fragment and tagged with adapter sequences using a transposase enzyme in a process known as tagmentation. Using a limited cycle PCR step, indices were introduced. This helped in demultiplexing after sequencing was complete. A size selection clean-up was done using AMPure beads (AGENCOURT) and normalized to ensure equal representation of all libraries. The normalized libraries were pooled, denatured and loaded on the Miseq platform with an out of 2X200bp [12].

## Bioinformatics analysis

Paired-end reads from each *E. coli* isolate were assembled de novo using spades v 3.11.1 algorithm [13] to generate a draft genome sequence for each isolate and quality assessment of for assemblies was done using QUAST 4.5. Clermont typing method [14, 15] was used to determine phylogroups basing on the genome-clustering tool called Mash [16].

Sequences were analysed using the Nullarbor pipeline (Seeman T, available at: https://github.com/tseemann/nullarbor). In brief, reads were trimmed to remove adaptor sequences and low-quality bases with Trimmomatic, [17] and Kraken (v1.1.1) was used to investigate for contamination [18]. Reads were aligned to the *E. coli* str. K-12 substr. MG1655 complete genome using the Burrows-Wheeler Aligner MEM (0.7.17-r1188) algorithm [19]. Samples with at least 21x depth of coverage and 76% genome coverage were retained for analysis. SNPs were identified using Freebayes (v1.3.1) with a minimum depth of coverage of 10x and allelic frequency of 0.9 required to confidently call a single nucleotide polymorphism (SNP) [20]. For all isolates, sequence reads covered >76% of the reference genome, A total of 29,525 core SNP sites were identified, with 2,733 genes identified as core genes (present in >99% of isolates).De novo assemblies were also performed using SPAdes (v. 3.13.1), [13] as part of the Nullarbor pipeline, with genes annotated using Prokka (v. 1.14.0, Seemann T, available at: https://github.com/tseemann/prokka).

*E. coli* has been traditionally clustered by phylogroup and have a good association with isolates being a commensal or pathogen [21]. The phylotyping method described by Clermont et al. (2013) [22] was performed *in silico*. In short, based on the presence or absence of 5 genes: *chuA*, *yjaA*, *tspE4.C2*, *arpA* and *trpA*, isolates can be separated into 8 phylogroups (A, B1, B2, D, C, E, F, and cryptic clades).

Each draft genome was screened for the presence of AMR and virulence genes using the software package Abricate (https://github.com/tseemann/abricate), a package for mass screening of contigs for antimicrobial resistance or virulence genes using databases. For antimicrobial resistance screening we used the NCBI Bacterial Antimicrobial Resistance Reference Gene Database (https://www.ncbi.nlm.nih.gov/bioproject/PRJNA313047) and for virulence genes screening VFDB, a reference database for bacterial virulence factors [23]. Statistical analysis and visualisation were performed in R version 3.6.1 (www.r-project.org).

## Ethical considerations

The study was approved by the Makerere University School of Biomedical Sciences Higher Degrees and Ethics Committee (SBS-HDREC) and The Uganda National Council of Science and Technology (UNCST). Written informed consent was obtained from all participants. All participant identifying information was kept confidential. Written informed consent from parents/guardians of participants below 18 years was sought and written assent was provided by all minors who were able to read and write who voluntarily participated in this study while those who were not able to read and write provided verbal assent. Verbal assent was witnessed by the parent/guardian.

## Results

### Antibiotic susceptibility pattern

Antibiotic susceptibility testing was carried out on all the 42 *E. coli* isolated from both human and animals. A majority of the isolates displayed high resistance to ampicillin, cefazolin, trimethoprim/sulphurmethoxazole, amoxicillin/clavulanic acid and Cefotaxime. Low resistance to imipenem, ciprofloxacin, Levofloxacin Ertapenem were recorded whereas only one isolate was resistant to gentamicin. Most of the isolates 41/42 were resistant to three or more antibiotics (multi-drug resistant) and 21/42 isolates were ESBL producers.

### Whole genome sequencing and de novo assembly

Computation of the total number of reads and quality metrics of the showed homogenous results with a good quality profile for all isolates assemblies. Estimated average depth of coverage for 42 *E. coli* isolates on which WGS was performed was 28× with an average of 197 contigs >500-base pairs after genome assembly using SPAdes.

### Phylogenetic analysis of the *E. coli* Isolated from humans and cattle

The genomes of *E. coli* from humans and animals were clustered according to single nucleotide polymorphism in core genes. At household level, the genomes from both human and animals clustered away from one another except for one instance where two human isolates from the same household clustered together. However, 67% of the *E. coli* isolated from cattle were closely related to those found in humans (Fig 1). All the four major *E. coli* phylotypes were identified through *in silico* phylotyping. Most isolates belonged to phylotype B1, which accounted for 38% (n = 16) of *E. coli* isolates, evenly split between human and cattle samples (Fig 1). Phylotypes A and D accounted for 31% (n = 13) and 9.5% (n = 4) of isolates respectively, with the majority of the phylogroup A isolated from human samples and only two from cattle. The majority of isolates from Hima location are phylotype B1 and A.

### Antibiotic susceptibility pattern

Antibiotic susceptibility testing was carried out on all the 42 *E. coli* isolated from both human and cattle. Generally, a majority of the isolates displayed high resistance to cefazolin (100%), ampicillin (98%), Cefotaxime (67%), trimethoprim/sulphurmethoxazole (64%) and amoxicillin/clavulanic acid (62%). Low resistance to ciprofloxacin (5%), imipenem (10%), and Levofloxacin (12%) were recorded whereas only one isolate was resistant to gentamicin (2%). Most of the isolates 41/42 were resistant to three or more antibiotics (multi-drug resistant) and 21/42 isolates were ESBL producers (Fig 1). All the isolates assigned to phylogroup A were resistant to ampicillin, cefazolin and trimethoprim/sulfamethoxazole but sensitive to imipenem, gentamicin, ciprofloxacin, levofloxacin and tetracycline. All phylogroup B1 isolates were resistant to cefazolin but sensitive to imipenem, gentamicin and tetracycline. There was also high resistance against ampicillin (94%), amoxicillin/clavulanic acid (94%).

Generally, there was a similar pattern of antibiotic resistance among the *E. coli* isolated from human and that isolated from cattle with the trends in cattle appearing higher than those in humans (Fig 2). All the isolates from cattle and humans were resistant to cefazolin. All the human isolates were resistant to ampicillin and highly resistant to trimethoprim/sulfamethoxazole 21(70%), amoxicillin/clavulanic acid, 20(67%) and Cefotaxime, 19(63%). Low resistance against gentamicin 1(3%), ciprofloxacin, 2(7%) and imipenem, 3(10%) was detected among the isolates from humans. The isolates from cattle were highly resistant to ampicillin, 11(91%) and Cefotaxime, 9(75%) and no resistance was detected against gentamicin, ciprofloxacin and

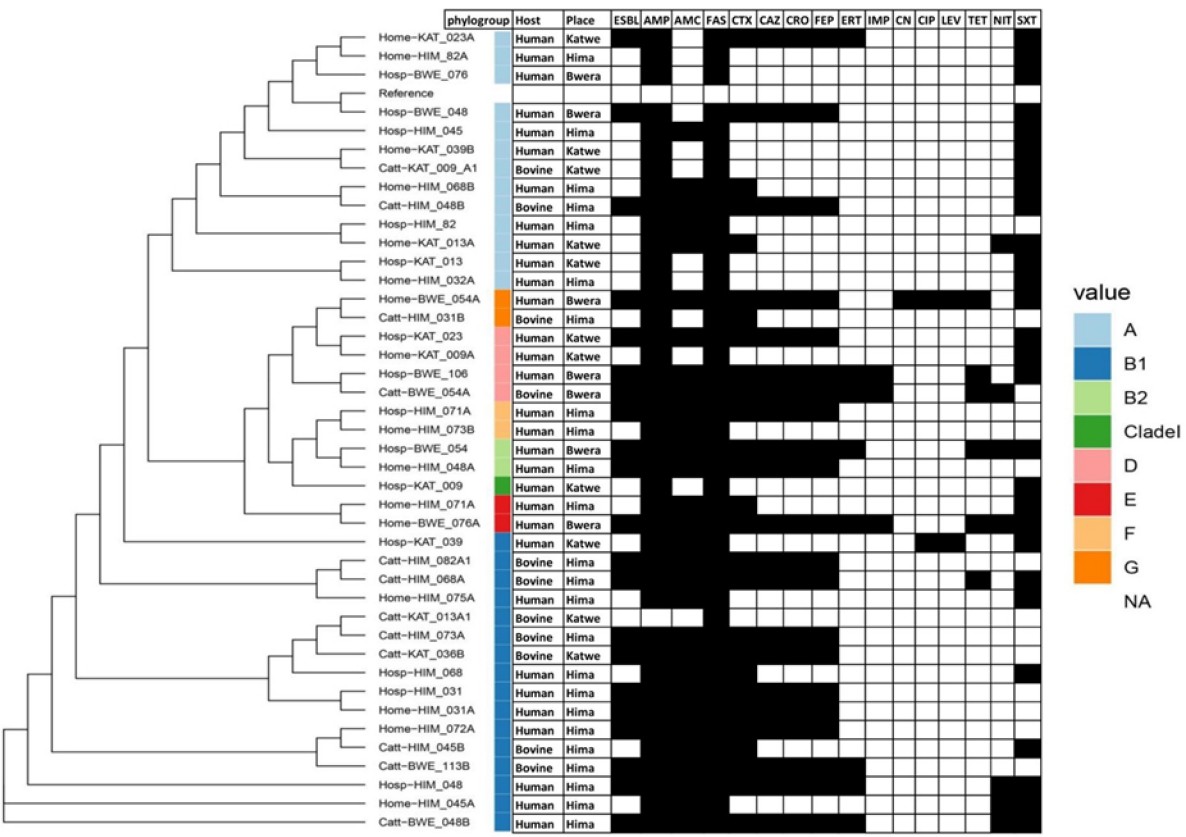

**Fig 1. Maximum likelihood phylogenetic trees based on SNP differences within the core genomes and antibiotic susceptibility of *E. coli* isolated from humans and animals in Kasese district.** The black shading indicates resistance. ESBL = Extended Spectrum β-lactamase, AMP = ampicillin, AMC = amoxicillin/clavulanic acid, FAS = cefazolin, CTX = Cefotaxime, CAZ = ceftazidime, CRO = ceftriaxone, FEP = cefepime, ERT = ertapenem, IMP = imipenem, CN = gentamicin CIP = ciprofloxacin, LEV = levofloxacin, TET = tetracycline, NIT = nitrofurantoin, SXT = trimethoprim/sulphurmethoxazole.

levofloxacin. There was no significant difference in resistance to particular drugs between the human and cattle isolates except for Cotrimoxazole (p<0.05).

## Comparison of virulence genes identified in *E. coli* genomes, isolated and sequenced from human and cattle samples

Antimicrobial resistance genes were identified in all 30 *E. coli* genomes from human and the 12 *E. coli* genomes from cattle. The carriage of virulence genes was significantly higher among the *E. coli* isolates from humans compared to those of cattle origin (Fig 3) and this rule out the possibility of clonal relationship between the isolates from the two hosts (p = 0.18). Isolates from Katwe location had significantly higher number of virulence genes compared to the other two sites, Bwera and Hima (p = 0.04). The human samples have a significantly higher number of virulence genes compared to bovine samples. However, there was no difference in the median of the virulence genes from the *E. coli* isolated from the hospital and that isolated from the community (Fig 4).

## Detection of virulence factors genes

A total of 174 different virulence genes were detected from the *E. coli* genomes. The virulence genes; cheY, csgG, entC, entF, entS, fepB, fepC, fepD and flgG were detected in all the *E. coli*

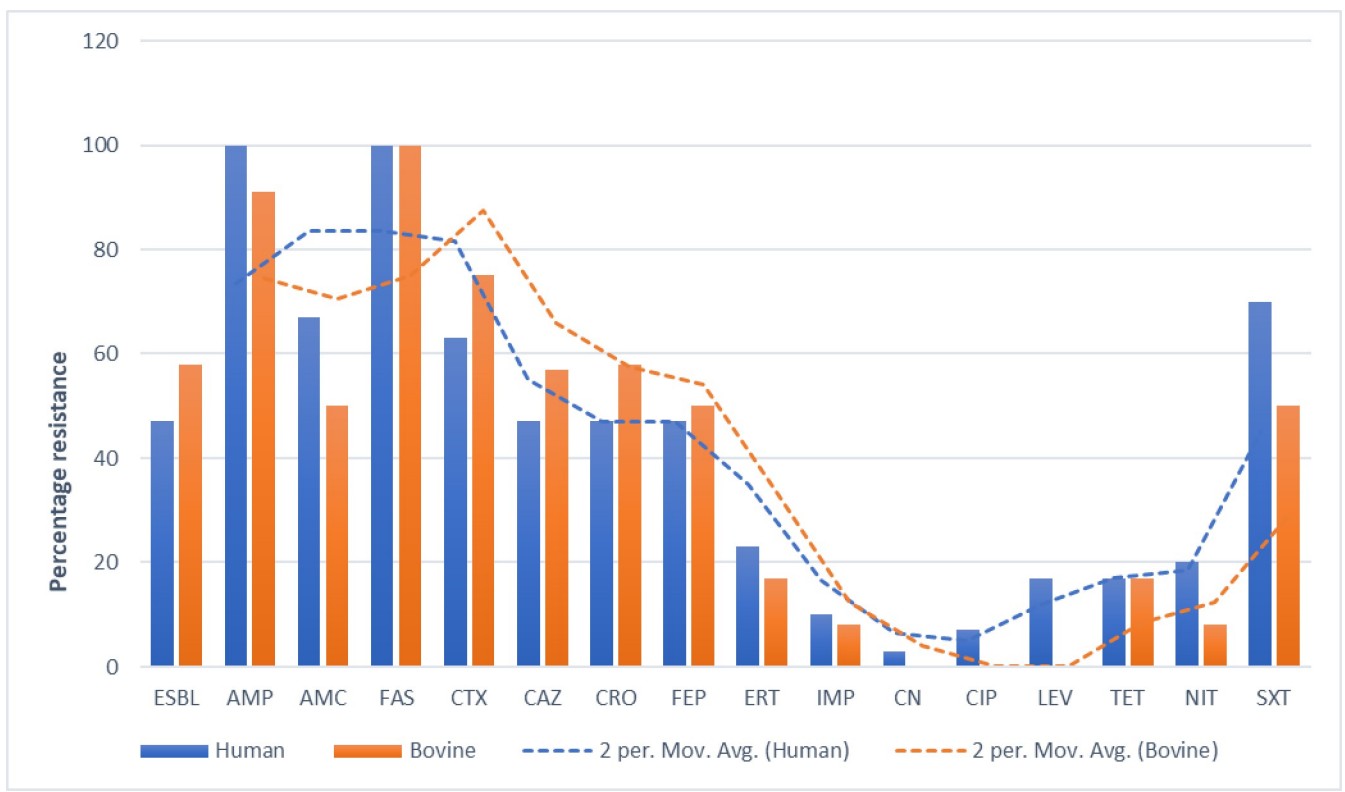

**Fig 2. Phenotypic antibiotic resistance patterns in humans and cattle.** ESBL = Extended Spectrum β-lactamase, AMP = ampicillin, AMC = amoxicillin/ clavulanic acid, FAS = cefazolin, CTX = Cefotaxime, CAZ = ceftazidime, CRO = ceftriaxone, FEP = cefepime, ERT = ertapenem, IMP = imipenem, CN = gentamicin CIP = ciprofloxacin, LEV = levofloxacin, TET = tetracycline, NIT = nitrofurantoin, SXT = trimethoprim/sulphurmethoxazole.

isolates. Other virulent genes that were commonly prevalent include; csgB, csgD, csgE, entA, entB, fdaC, fimA, fim B, fimC, fimD, fimE, fimF, fimG, fimH, fimI, flgG, flhA, fliG, fliM, flip, gspC, gspD, gspE, gspF, gspG, gspH, gspI, gspJ, gspK, gspL, gspM, ompA, yagV/ecpE, yagW/ ecpD, yagX/ecpC, yagY/ecpB, yagZ/ecpA and ykgk/ecpB. The proportion of virulence genes from the two hosts was analyzed and presented in box plots (Fig 3).

## Discussion

Whole genome sequencing was used to analyse non–duplicate *E. coli* isolated from both human and cattle in pastoralist communities of Kasese district. Phylogenetic analysis showed that the genomes of the human *E. coli* generally clustered together and away from those of cattle origin and likewise the genomes from cattle tended to cluster together and away from those of the cattle origin. The human isolates were mainly assigned to phylogroup A and B1 whereas the cattle isolates were mostly assigned to phylogroup B1. This finding agrees with the Zambian study which reported that most of the cattle isolates of E. coli were assigned to phylogroup B1 which is commonly associated with commensal existence [24]. Phylogroups B2, Clade l, E and F; were associated with only the *E. coli* isolated from humans, three human isolates were assigned to phylogroup D and only one cattle isolate was assigned to group D. The presence of phylogroups B2 and D among human isolates presents a risk for infection due to *E. coli* in humans. Phylogroups B2 and D have been reported to be pathogenic and responsible for extra-intestinal infections by a number of studies [25–28].

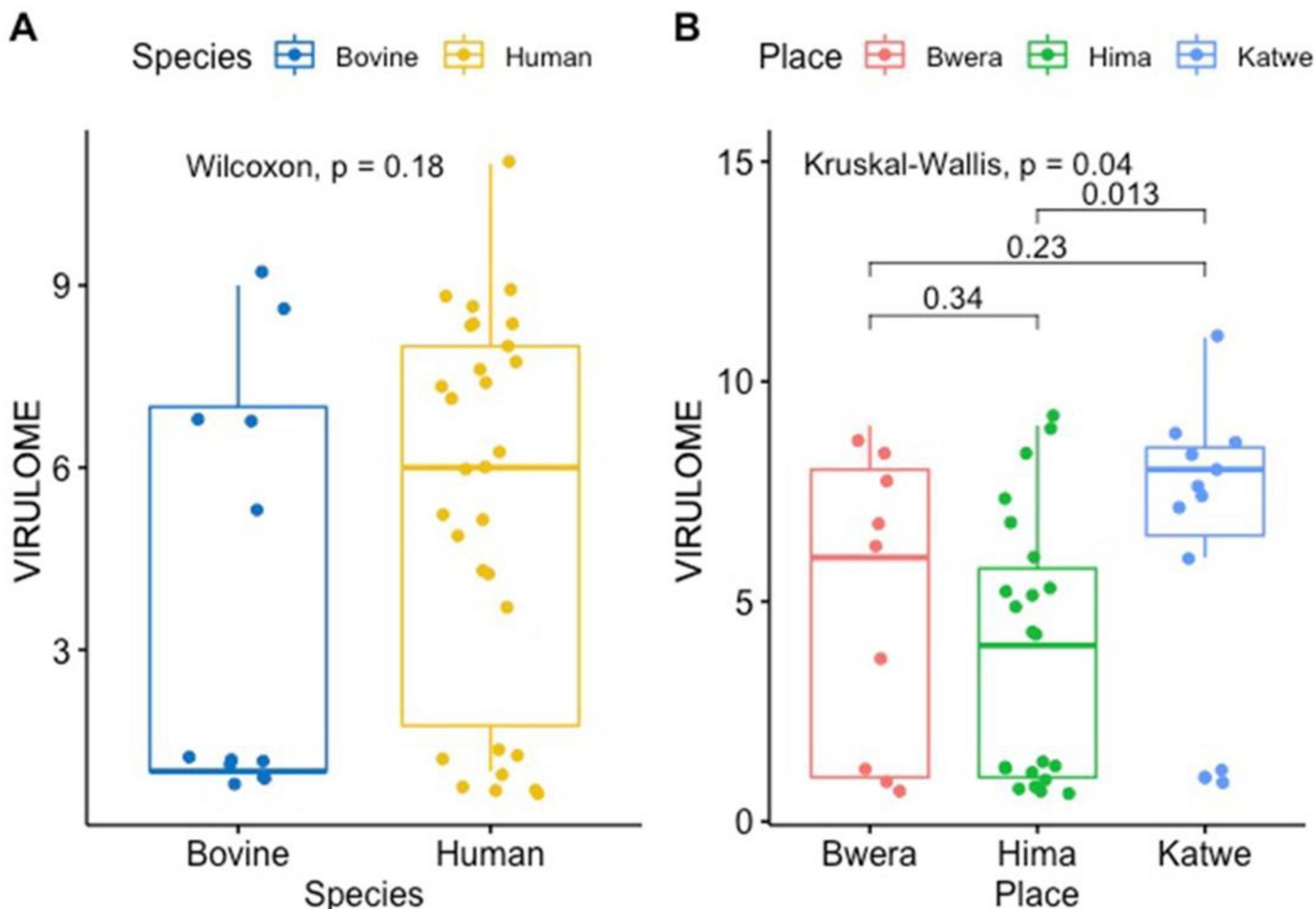

**Fig 3. Box plots showing the median values for the number of virulence genes in the two groups along with the interquartile range.** The human samples had a significantly higher number of virulence genes compared to bovine samples. The *E. coli* strains from Katwe had significantly higher number of virulence genes compared to Bwera and Hima.

Phenotypic resistance testing revealed a similar resistance pattern among the human isolates than the cattle isolates for the 15 antibiotics tested in this study. A similar study in Lusaka-Zambia demonstrated significantly higher levels of resistance among the human *E. coli* isolates when the same antibiotics were tested [24]. Every year, humans produce, use, and consume approximately 175,000 tons of antibiotics [29] which makes emergency of AMR in humans, animals and environment inevitable. The resistance genes were also compared in *E. coli* isolates originating from the two hosts. The carriage of multiple AMR genes among the *E. coli* population from humans was higher than those from cattle (Fig 5). There have been several debates trying to link the emergency of AMR in humans to antibiotic use in food animals [30–32]. Our study, like the study in Zambia [24], showed that there was much more diversity in the resistance genotypes present in human isolates than in the cattle isolates. The deference in the diversity of the resistance genotype could indicate the existence of independent AMR selection pressures in human and animals. Whereas other studies have demonstrated the predominance of blaCTX-M-15 allele among human and animal isolates of *E. coli*, from the community [33, 34], this study identified arsB-mob (80%) as the most predominant allele (Fig 2).

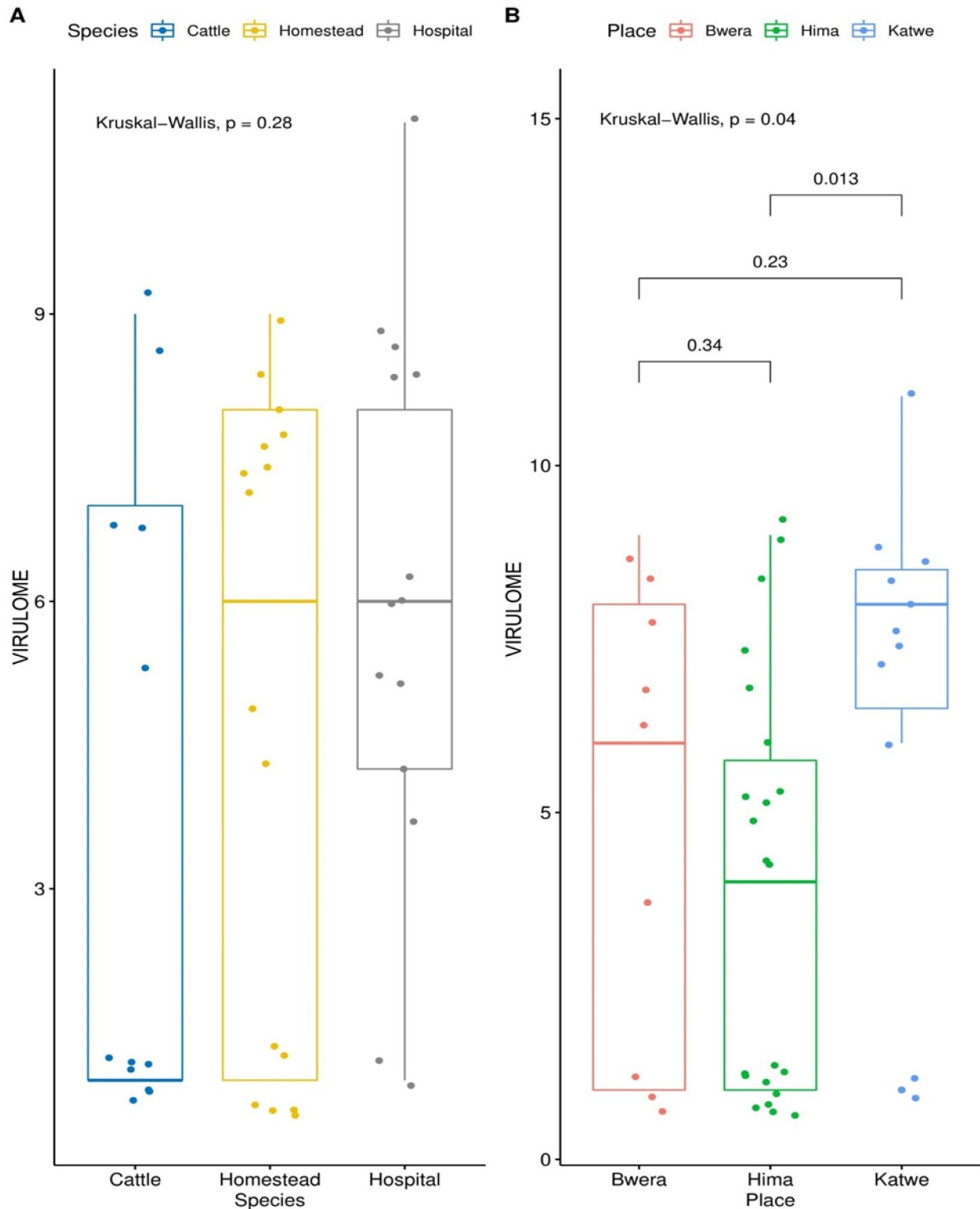

**Fig 4. Box plots comparing the variation in virulence of the *E. coli* from hospital and the community.** Homestead = community isolated *E. coli*, species = human/cattle.

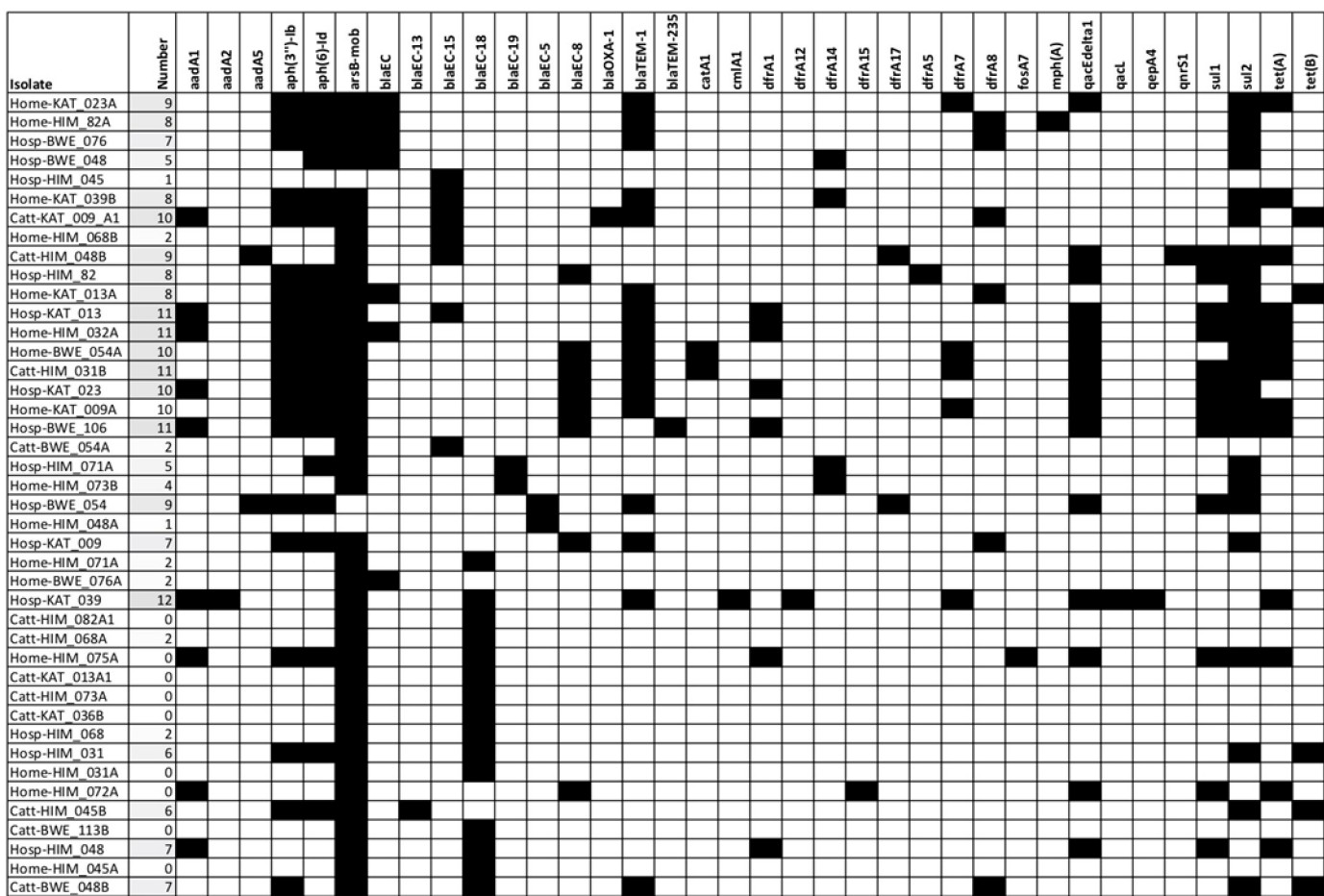

**Fig 5. Heat map showing AMR genes found in the genomes of sequenced *E. coli* isolates.** AMR genes are annotated with the NCBI Bacterial Antimicrobial Resistance Reference Gene Database with >95% coverage. The names of the strains indicated on the y axis are presented in the same order as in Fig 2. aadA1, aadAd2, aadAD5: Streptomycin 3''-adenylyltransferase; aph(3'')-Ib, aph(6)-Id: Streptomycin phosphotransferase; arsB-mob: Arylsulfatase B; blaEC, blaEC-13, blaEC-15, blaEC-18, blaEC-19, blaEC-5, blaEC-8: Beta-lactamase; blaOXA-1: Class D beta-lactamases; blaTEM-1, blaTEM-235: Beta-lactamase; catA1: chloramphenicol acetyl transferase; cmlA1: chloramphenicol efflux transporter; dfrA1, dfrA12, dfrA14, dfrA15, dfrA17, dfrA5, dfrA7, dfrA8: trimethoprim resistant dihydrofolate reductase; fosA: fosfomycin thiol transferase; mph(A): macrolide phosphotransferase; oqxB11: RND efflux pump conferring resistance to fluoroquinolone; qacL, qacEdelta1: Small multidrug resistance gene; qepA4: quinolone efflux pump; qnrS1: quinolone resistance gene; sul1, sul2, sul3: sulfonamide resistant; tet(A), tet(B): tetracycline efflux pump (Fig 5).

This discrepancy could be due to the different methods used and undetermined geographical factors. Another study in livestock in Uganda indicated that blaCTX-M and carbapenems resistant genes were not harboured by *E. coli* from livestock [35]. The prevalence of cephalosporin resistance genes in the *E. coli* isolated from cattle in this study presents a risk of transmission of these resistant genes from animals to humans. Similar observations have been made by Okubo et al., in Uganda [35]. In this study, the prevalence of resistance genes was high both in human and cattle isolates.

A high variation of virulence genes was registered among the *E. coli* genome from humans than those of cattle origin further supports the hypothesis that the *E. coli* isolated from the two hosts have a common ancestral origin. However, some studies have found clonal commonality between *E.coli* isolated from the hospital, the community and animals [36]. The reason for this discrepancy is mainly due to the different methods used. Also, there was no difference in median of virulence genes between the *E. coli* isolated from hospital from that isolated from

the community and this may imply that the *E. coli* isolated from the two hosts have similar evolutionary pressures.

## Conclusions

From the analysis of the core genome and phenotypic resistance, this study has demonstrated that the *E. coli* of human origin and those of cattle origin may have a common ancestry. Limited sharing of virulence genes presents a challenge to the notion that AMR in humans is as a result of antibiotic use in the farm and distorts the picture of the directionality of transmission of AMR at a human-animal interface and presents a task of exploring alternative routes of transmission of AMR. A major limitation of this study is the small number of isolates sequenced and only one species of animals studied, therefore, we recommend a larger study for better generalization of results.

## Acknowledgments

We gratefully acknowledge our participants for making this study possible. Heartfelt thanks to Stallone Kisembo and Zaydah De Laurent for the great work they did as research assistants.

## Author Contributions

**Conceptualization:** Jacob Stanley Iramiot, Benon B. Asiimwe.

**Data curation:** Jacob Stanley Iramiot, Benon B. Asiimwe.

**Formal analysis:** Jacob Stanley Iramiot, Etienne P. de Villiers.

**Funding acquisition:** Benon B. Asiimwe.

**Investigation:** Jacob Stanley Iramiot.

**Methodology:** Jacob Stanley Iramiot.

**Project administration:** Jacob Stanley Iramiot, Benon B. Asiimwe.

**Resources:** Benon B. Asiimwe.

**Software:** Etienne P. de Villiers.

**Supervision:** Henry Kajumbula, Joel Bazira, Etienne P. de Villiers, Benon B. Asiimwe.

**Validation:** Henry Kajumbula, Joel Bazira, Etienne P. de Villiers.

**Visualization:** Etienne P. de Villiers.

**Writing – original draft:** Jacob Stanley Iramiot.

**Writing – review & editing:** Jacob Stanley Iramiot, Henry Kajumbula, Joel Bazira, Etienne P. de Villiers, Benon B. Asiimwe.

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
