## [Decision Letter · Decision Letter 0]

16 Apr 2020

PONE-D-20-09327

Whole genome sequences of multi-drug resistant Escherichia coli isolated in a Pastoralist Community of Western Uganda: Phylogenomic changes, virulence and resistant genes

PLOS ONE

Dear Mr Iramiot,

Thank you for submitting your manuscript to PLOS ONE. After careful consideration, we feel that it has merit but does not fully meet PLOS ONE’s publication criteria as it currently stands. Therefore, we invite you to submit a revised version of the manuscript that addresses the points raised during the review process.

We would appreciate receiving your revised manuscript by May 31 2020 11:59PM. To enhance the reproducibility of your results, we recommend that if applicable you deposit your laboratory protocols in protocols.io, where a protocol can be assigned its own identifier (DOI) such that it can be cited independently in the future. For instructions see: http://journals.plos.org/plosone/s/submission-guidelines#loc-laboratory-protocols

We look forward to receiving your revised manuscript.

Kind regards,

Grzegorz Woźniakowski, PhD ScD

Academic Editor

PLOS ONE

2. Please provide additional details regarding participant consent. In the ethics statement in the Methods and online submission information, please ensure that you have specified how verbal consent was documented and witnessed.

4. Your ethics statement must appear in the Methods section of your manuscript. If your ethics statement is written in any section besides the Methods, please move it to the Methods section and delete it from any other section. Please also ensure that your ethics statement is included in your manuscript, as the ethics section of your online submission will not be published alongside your manuscript.

5. Please ensure that you refer to Figure 5 in your text as, if accepted, production will need this reference to link the reader to the figure.

6. We note that Figure 1 in your submission contain map images which may be copyrighted. All PLOS content is published under the Creative Commons Attribution License (CC BY 4.0), which means that the manuscript, images, and Supporting Information files will be freely available online, and any third party is permitted to access, download, copy, distribute, and use these materials in any way, even commercially, with proper attribution. For these reasons, we cannot publish previously copyrighted maps or satellite images created using proprietary data, such as Google software (Google Maps, Street View, and Earth). For more information, see our copyright guidelines: http://journals.plos.org/plosone/s/licenses-and-copyright.

Reviewers' comments:

Reviewer's Responses to Questions

**Comments to the Author**

1. Is the manuscript technically sound, and do the data support the conclusions?

Reviewer #1: Partly

Reviewer #2: Yes

2. Has the statistical analysis been performed appropriately and rigorously? 

Reviewer #1: Yes

Reviewer #2: Yes

3. Have the authors made all data underlying the findings in their manuscript fully available?

Reviewer #1: Yes

Reviewer #2: Yes

4. Is the manuscript presented in an intelligible fashion and written in standard English?

Reviewer #1: No

Reviewer #2: Yes

5. Review Comments to the Author

Reviewer #1: The study provided by Authors gives an overview on genetic profile of collected strains (n=42) from cattle and human in Kasese District, Uganda. The Authors concluded that major results stays contrary to thesis that antimicrobial resistance (AMR) may be passed to human from livestock as a result of antibiotic use in the farm.

Methods provided by Authors are sound and well based. However discussion section and conclusion should be revised, please find following remarks below:

A) Major remarks:

1. Please highlight the exact number of tested strains in abstract. Cattle (n=12) and human (n=30). In abstract we can found: […] 21/42 isolates were ESBL producers; 13/42 from human and 8/42 from cattle – it remains unclear, and suggest there were 84 samples. I suggest to use 13/30 and 8/12 instead, respectively.

2. Please indicate the exact number of tested strains in Methods section, subsection Bacterial strains as mentioned above

3. Major conclusion made by Authors stays a little bit contrary to data provided in Figure 2. According to Figure 2 about 8 cattle E. coli strains (60% of all cattle samples) are closely related to this found in human and the several of them have similar or wider pattern of AMR.

4. Could you explain why only cattle was designated to the study? If the cattle is the most important meat donor in Uganda, such thing should be mentioned in the background section.

5. Consider discuss other work i.e work of Okubo et al (1) in meaning possibility of obtaining AMR genes from other livestock

6. Because of relative small number of samples in case of cattle (n=12), and no comparison to other livestock (lack of discussion i.e abovementioned) the conclusion may be misleading.

B) Minor remarks:

1. consider the necessity of Figure 1. It is not essential for study.

- Please correct:

2. Method section: Whole genome-sequencing: “Briefly, gDNA was fragmented and

tagged with […]”

3. Figure 2: last strain: Catt-BWE_048B – should be probably Bovine not Human in Host section.

4. I suggest to revise paper by native English speaker.

In conclusion, the study provided by authors are interesting and presents valuable data. I recommend to publish it after revision.

1. Okubo T, Yossapol M, Maruyama F, Wampande EM, Kakooza S, Ohya K, et al. Phenotypic and genotypic analyses of antimicrobial resistant bacteria in livestock in Uganda. Transbound Emerg Dis [Internet]. 2019 Jan 22;66(1):317–26. Available from: https://doi.org/10.1111/tbed.13024

Reviewer #2: The article prepare by the authors present a serious problem which is E. coli multi-drug resistance. It is very interesting that it was observed similar pattern of antibiotic resistance both in human and cattle. Paper is written correctly, the Figures are very impressing, however there are small details which should be corrected.

In Manuscript draft there are missing keywords, however there are present in main document so I assumed that this is omission. In some places like in page 5 Escherichia coli is not written in italic. When you once use shortcut "E. coli" in the text do not use full name in further part. You should write "ng/µl" not "ng/ul" (page 4).

In Results I thought that it will be more genetic data, knowing the title of article. That part could be improved. In addition despite the fact that the Discussion is written correctly and in scientific way, it also could be improved. For a research article the Discussion is to short, try write more about importance of your results regarding to other papers.

Summary, the article is very interesting and the data are important equally for human health and agriculture. I recommend this paper for publication after minor revision.

6. PLOS authors have the option to publish the peer review history of their article (what does this mean?). If published, this will include your full peer review and any attached files.

Reviewer #1: No

Reviewer #2: No

---

## [Author Response · Author response to Decision Letter 0]

21 Apr 2020

Response to Reviewers

Editor’s comments 

SN COMMENT AUTHOR’S RESPONSE 

1 Please ensure that your manuscript meets PLOS ONE's style requirements, including those for file naming. This has been corrected as recommended 

2 Please provide additional details regarding participant consent. In the ethics statement in the Methods and online submission information, please ensure that you have specified how verbal consent was documented and witnessed. Additional details regarding participant consent and the ethical statement 

3 We note that you have stated that you will provide repository information for your data at acceptance. Should your manuscript be accepted for publication, we will hold it until you provide the relevant accession numbers or DOIs necessary to access your data. If you wish to make changes to your Data Availability statement, please describe these changes in your cover letter and we will update your Data Availability statement to reflect the information you provide. The data availability statement has been updated.

4 Your ethics statement must appear in the Methods section of your manuscript. If your ethics statement is written in any section besides the Methods, please move it to the Methods section and delete it from any other section. Please also ensure that your ethics statement is included in your manuscript, as the ethics section of your online submission will not be published alongside your manuscript The ethical statement has been relocated to the methods

5 Please ensure that you refer to Figure 5 in your text as, if accepted, production will need this reference to link the reader to the figure Figure 5 has been referenced in the text

6 Please either i) remove the figure or ii) supply a replacement figure that complies with the CC BY 4.0 license. The figure has been removed 

Reviewer 1

SN COMMENT AUTHOR’S RESPONSE

1 Please highlight the exact number of tested strains in abstract. Cattle (n=12) and human (n=30). In abstract we can found: […] 21/42 isolates were ESBL producers; 13/42 from human and 8/42 from cattle – it remains unclear, and suggest there were 84 samples. I suggest to use 13/30 and 8/12 instead, respectively. The exact numbers tested have been highlighted as recommended by the reviewer. Thank you

2 Please indicate the exact number of tested strains in Methods section, subsection Bacterial strains as mentioned above Amended as recommended by the reviewer 

3 Major conclusion made by Authors stays a little bit contrary to data provided in Figure 2. According to Figure 2 about 8 cattle E. coli strains (60% of all cattle samples) are closely related to this found in human and the several of them have similar or wider pattern of AMR. The conclusion has been amended as to reflect the results.

4 Could you explain why only cattle was designated to the study? If the cattle is the most important meat donor in Uganda, such thing should be mentioned in the background section. The pastoralists in Kasese district majorly reared cattle. This has been mention in the manuscript

5 Consider discuss other work i.e work of Okubo et al (1) in meaning possibility of obtaining AMR genes from other livestock This has been used to enrich the discussion, thank you.

6 Because of relative small number of samples in case of cattle (n=12), and no comparison to other livestock (lack of discussion i.e abovementioned) the conclusion may be misleading. This has acknowledged as a limitation

7 consider the necessity of Figure 1. It is not essential for study. The figure has been deleted

8 Method section: Whole genome-sequencing: “Briefly, gDNA was fragmented and

tagged with […]” The reference has been added

Reviewer 2

SN COMMENT AUTHOR’S RESPONSE

1 In Manuscript draft there are missing keywords, however there are present in main document so I assumed that this is omission. In some places like in page 5 Escherichia coli is not written in italic. When you once use shortcut "E. coli" in the text do not use full name in further part. You should write "ng/µl" not "ng/ul" (page 4). Key words have been added and all other comments addressed as recommended by the reviewer 

2 In Results I thought that it will be more genetic data, knowing the title of article. That part could be improved. In addition despite the fact that the Discussion is written correctly and in scientific way, it also could be improved. For a research article the Discussion is to short, try write more about importance of your results regarding to other papers. Some adjustments have been made

---

## [Decision Letter · Decision Letter 1]

12 May 2020

Whole genome sequences of multi-drug resistant Escherichia coli isolated in a Pastoralist Community of Western Uganda: Phylogenomic changes, virulence and resistant genes

PONE-D-20-09327R1

Dear Dr. Iramiot,

We are pleased to inform you that your manuscript has been judged scientifically suitable for publication and will be formally accepted for publication once it complies with all outstanding technical requirements.

With kind regards,

Grzegorz Woźniakowski, PhD ScD

Academic Editor

PLOS ONE

Additional Editor Comments (optional):

Reviewers' comments:

Reviewer's Responses to Questions

**Comments to the Author**

1. If the authors have adequately addressed your comments raised in a previous round of review and you feel that this manuscript is now acceptable for publication, you may indicate that here to bypass the “Comments to the Author” section, enter your conflict of interest statement in the “Confidential to Editor” section, and submit your "Accept" recommendation.

Reviewer #1: All comments have been addressed

Reviewer #2: All comments have been addressed

2. Is the manuscript technically sound, and do the data support the conclusions?

Reviewer #1: Yes

Reviewer #2: (No Response)

3. Has the statistical analysis been performed appropriately and rigorously? 

Reviewer #1: Yes

Reviewer #2: (No Response)

4. Have the authors made all data underlying the findings in their manuscript fully available?

Reviewer #1: Yes

Reviewer #2: (No Response)

5. Is the manuscript presented in an intelligible fashion and written in standard English?

Reviewer #1: No

Reviewer #2: (No Response)

6. Review Comments to the Author

Reviewer #1: Dear Author

Thank you for applying indicated remarks.

Please verify english by native speaker.

I've got no further comments.

I recommend to publish manuscript after english verification.

Kind regards.

Reviewer #2: (No Response)

7. PLOS authors have the option to publish the peer review history of their article (what does this mean?). If published, this will include your full peer review and any attached files.

Reviewer #1: No

Reviewer #2: No

---

## [Editor Report · Acceptance letter]

15 May 2020

PONE-D-20-09327R1 

Whole genome sequences of multi-drug resistant *Escherichia coli* isolated in a Pastoralist Community of Western Uganda: Phylogenomic changes, virulence and resistant genes 

Dear Dr. Iramiot:

I am pleased to inform you that your manuscript has been deemed suitable for publication in PLOS ONE. Congratulations! Your manuscript is now with our production department. 

With kind regards,

on behalf of

Prof. Grzegorz Woźniakowski 

Academic Editor

PLOS ONE